# Genetic diversity and population structure of *Phlebotomus argentipes*: Vector of *Leishmania donovani* in Sri Lanka

**Dulani Ruwanika K. Pathirage**[1], **Thilini C. Weeraratne**[2], **Sanath C. Senanayake**[1], **S. H. P. Parakrama Karunaratne**[2‡], **Nadira D. Karunaweera**[1‡*]

**1** Faculty of Medicine, Department of Parasitology, University of Colombo, Colombo, Sri Lanka, **2** Faculty of Science, Department of Zoology, University of Peradeniya, Peradeniya, Sri Lanka

‡ Co-principal Investigators of this work
* nadira@parasit.cmb.ac.lk

**Data Availability Statement:** All relevant relevant data have been uploaded to GenBank: cox 1-

## Abstract

*Phlebotomus argentipes* is the vector of *Leishmania donovani* which causes the disease leishmaniasis, a neglected tropical disease and a growing health problem in Sri Lanka. A proper understanding of the population genetic structure of sand fly vectors is considered important prior to planning and implementation of a successful vector control program. Thus, the present study was conducted to determine the population genetic structure of sand fly vectors in Sri Lanka. Two mitochondrial genes namely *Cytochrome c oxidase subunit 1* (*Cox 1*) and *Cytochrome b* (*Cytb*), and the internal transcribed spacer 2 (ITS2) region from the nuclear ribosomal DNA were used for molecular characterization. Analyses included maximum likelihood method, network analysis and DNA polymorphisms. The outcome revealed unique sequences of all genomic regions studied except the *cox 1* gene had a relationship with sand flies isolated previously from Sri Lanka, India and Israel and *cytb* gene of 4 sand flies that aligned with those isolated earlier from Sri Lanka and 3 from Madagascar. Furthermore, *cox 1* gene and ITS 2 region analyses based on $F_{ST}$ values indicated a possible gene flow between the study sites whereas *cytb* gene analysis favoured the existence of genetically distinct populations of *P. argentipes* in each of the study sites. Poor population differentiation of *P. argentipes*, a possible consequence of a gene flow, is indeed of concern due to the risk imposed by promoting the spread of functionally important phenotypes such as insecticide resistance across the country, making future vector control efforts challenging.

## Introduction

Cutaneous leishmaniasis (CL), muco-cutaneous leishmaniasis (MCL) and visceral leishmaniasis (VL) or "kala-azar" are the three main clinical forms of the disease leishmaniasis [1]. The disease is caused by an obligate intracellular flagellated protozoan belonging to the genus *Leishmania*. *Phlebotomus argentipes* (Diptera: Psycodidae) is the known vector of *Leishmania donovani*, the

MW256437-MW256478 cytb- MW571044-
MW571085 ITS2- MW322926-MW322967.

**Funding:** This work was supported by the National Institute of Allergy and Infectious Diseases of the National Institutes of Health, USA, under award number U01AI136033. The content is solely the responsibility of the authors and does not necessarily represent the official views of the National Institutes of Health.

**Competing interests:** The authors have declared that no competing interests exist.

causal agent of the clinical form VL which is considered the second parasitic disease cause of a high number of deaths, in India, Nepal and Bangladesh [2–4]. Thus VL is recognized as the most virulent form among the three types of the disease [4]. However, the majority of *L. donovani* infections in Sri Lanka manifest as CL with only a few MCL and VL cases [5]. Although it was an exclusively imported disease prior to 1990s [6], now the disease is considered as widely prevalent over the country with case numbers increasing since 2001 and considered 'notifiable' in the health sector [1,5]. No national control programme is yet in place to contain the situation although leishmaniasis is a rapidly growing health threat in Sri Lanka [1].

Use of insecticides is the main strategy used for vector control in Sri Lanka, which may have promoted the development of insecticide resistance to indoor residual spraying of DDT and malathion [7]. This situation necessitated the use of organophosphates (e.g. fenitrothion), pyrethroids such as λ-cyhalothrin, cyfluthrin, and deltamethrin or the pseudo-pyrethroid etofenprox since 1994–2012 [8]. Rotation of insecticides every 3–5 years and sprayings restricted to targeted areas are used to overcome the development of resistance by the Sri Lankan malaria vector control programme. Sand flies were not directly targeted however, those flies in malaria endemic areas in the dry zone, which covers two-thirds of the country may have been regularly exposed to insecticides [9]. The presence of *kdr* mutation which is located at the position of 1014 with the amino acid changes from leucine to phenylalanine in a notable proportion of flies in Sri Lanka [10] will spread from one population to another by means of gene flow in sand flies similar to the case of mosquitoes [11–14].

The genetic structure for this sand fly species has been previously described [15,16]. However, the diversity of the phlebotomine sand flies is believed to be high in the country and the taxonomy based on morphology makes it hard to discriminate species and sibling species, which is more abundant among several species of phlebotomine sand flies.

DNA barcoding is considered as an efficient tool for genetic characterization of many species of insects including the New world sand flies and Old world sand flies by the use of *Cytochrome oxidase subunit I* (*cox 1*) sequences [16–19]. The 18s, 28s rDNA and *cox 1*, internal transcribed spacer 2 (ITS2) and *Cytochrome b* (*cytb*) sequences have been used to confirm the identification of species *Phlebotomus argentipes* complex in Sri Lanka. *Cytochrome oxidase subunit I* and *cytb* sequences were variable within the *P. argentipes* complex while the 18s and 28s sequences have not exhibited any variation within the sibling species [15].

The *cox 1*, *cytb* and ITS2 genes are useful genetic markers that are widely used in investigating the genetic characterization of Phlebotominae sand flies to determine the haplotypes, phylogeography and genetic structure [20,21].

The primary goal of this study was to use mitochondrial and ribosomal DNA sequence information to construct a DNA barcode to study the effect of evolutionary forces that may spread resistance genes from one population to another.

## Materials and methods

### Collection of sand flies

Adult sand flies were collected for two days at a time, every two months for a period of two years from November 2015 to November 2017 covering the sites Talawa (8˚14'11.468"N, 80˚21'2.782"E) in Anuradhapura district (North-Central Province), Pannala (7˚19'43.608"N, 80˚1'26.3316"E) in Kurunegala district (North-Western Province), Mamadala (6˚07'16.80"N, 81˚07'12.60"E) in Hambantota district (Southern Province), and Mirigama (7˚13'30.72"N, 80˚7'40.439"E) in Gampaha district (Western Province) in Sri Lanka (Fig 1). As the country is arbitrarily divided into dry, intermediate and wet zones based on the annual rainfall, Talawa and Mamadala are located within the dry zone, Pannala in the intermediate zone and Mirigama in

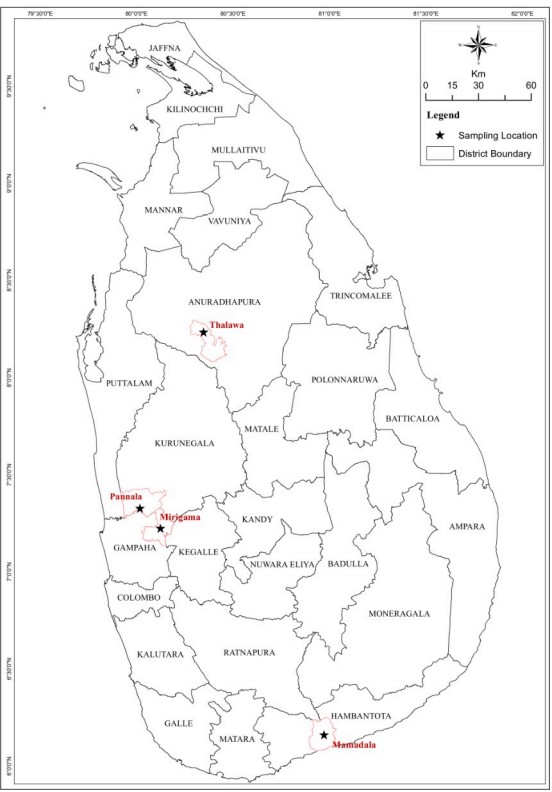

**Fig 1. Map of Sri Lanka.** The locations of the sand fly collection sites are indicated. Reprinted from authors' previous work [Pathirage *et al.* 2020 [10]].

the wet zone (Fig 1). The collection locations were determined based on climatic zones in the country, leishmaniasis case burden [22] and vector prevalence data [23]. The standard cattle-baited net traps and Center for Disease Control (CDC) light traps were used for overnight sand fly collections in each study site and flies were collected each morning at 6:00 h.

## Morphological identification

Identification of individual sand flies prior to PCR amplification was carried out using morphometric characteristics, up to the species level by examination under a light microscope at a magnification of 400× using standard taxonomic keys [24,25]. Identified samples were preserved in −20˚C [16] for PCR amplification.

## PCR amplification and sequencing of mitochondrial DNA (mtDNA) and ribosomal DNA (rDNA)

Full genomic DNA was isolated from individual sand flies (minimum of 10 individuals per population) collected from different study sites using DNeasy Blood and Tissue kit (QIAGEN-Germany).

PCR amplification of a variable part of the *cytochrome oxidase subunit I* gene was done using primers LCO 1490 (5′-GGTCAACAAATCATAAAGATATTGG-3′) and HCO 2198 (5′-TAA ACTTCAGGGTGACCAAAAAATCA-3′) [18] in a 50 μl PCR reaction which contained 0.4 μl of 10 μm of each primer, 10 μl of 5X Colourless GoTaq® Flexi Reaction Buffer (Promega, USA), 2 μl of 25 mM MgCl$_2$, 1 μl of 10 mM of each dNTP and 5 units of GoTaq® Flexi DNA

polymerase (Promega, USA) with the following conditions: an initial denaturation at 95˚ C for 5 min; followed by 5 cycles at 94˚ C for 40 s, 45˚ C for 1 min and 72˚ C for 1 min; 35 cycles at 94˚ C for 40 s, 51˚ C for 1 min, 72˚ C for 1 min; and a final extension at 72˚ C for 7 min.

The *cytochrome oxidase b* gene was amplified using primers CB3: 5 ′– CA [T/C] ATT CAA CC[A/T]GAATGATA–3 ′; N1N-PDR: 5 ′ – GGTA [C/T] [A/T] TTGC CTCGA [T/A] TTCG [T/A]TATGA– 3′ [26] in a 50 µl PCR reaction which contained 0.6 µl of 10 µm of each primer, 10 µl of 5X Colourless GoTaq® Flexi Reaction Buffer (Promega, USA), 3 µl of 25 mM MgCl₂, 1 µl of 10 mM of each dNTP and 5 units of GoTaq® Flexi DNA polymerase (Promega, USA). The *cytb* gene was amplified using the following conditions: an initial denaturation at 95˚ C for 5 min; followed by 10 cycles at 94˚ C for 30 s, 40˚ C for 30 s and 72˚ C for 1.5 min; 30 cycles at 94˚ C for 30 s, 45˚ C for 30 s, 72˚ C for 1.5 min; and a final extension at 72˚ C for 10 min.

Universal ITS2 primers forward primer (5.8S): 5′– ATC ACT CGC CTC ATG GAT CG 3′; reverse primer (28S): 5′– ATG CTT AAA TTT AGG GGG TAG TC 3′ [15] were used to amplify the ITS2 region. PCR amplification was done in a 25 µl PCR reaction which contained 0.820 µl of 10 µm of 5.8s and 0.730 µl of 2.8s each primer, 5 µl of 5X Colourless GoTaq® Flexi Reaction Buffer (Promega, USA), 1.5 µl of 25 mM MgCl₂, 0.5 µl of 10 mM of each dNTP and 5 units of GoTaq® Flexi DNA polymerase (Promega, USA) with cycling conditions of an initial denaturation at 95˚ C for 5 min; followed by 35 cycles at 95˚ C for 30 s, 55˚ C for 45 s and 72˚ C for 1 min; and a final extension at 72˚ C for 10 min.

PCR products were electrophoresed on a 1% agarose gel stained with ethidium bromide. Amplified PCR products were sent to Macrogen, Korea for purification and DNA sequencing using an Applied Biosystems 3730 DNA Analyzer. Sequences were received for both forward and reverse directions.

**Sequence analysis.**   Trace files of DNA sequences were manually edited and aligned using ClustalW in BioEdit 7.2.5 software. The finalized sequences were subjected to a BLAST search, on the National Center for Biotechnology information (NCBI) database GenBank (http://www.ncbi.nlm.nih.gov) for species confirmation. Sequences obtained from each genetic marker was deposited in Genbank database (Accession numbers: *cox 1*-MW256437-MW256478, *cytb*- MW571044- MW571085 and ITS2- MW322926-MW322967). Phylogenetic analysis was conducted using MEGA version 6.06 software (http://www.megasoftware.net/) and the trees were constructed using the Kimura-2 parameter distance model in maximum likelihood method [27] for *Phlebotomus argentipes* sensu lato complex in Sri Lanka. Number of haplotypes (h), genetic diversity indices [Haplotype Diversity Index (Hd) and Nucleotide Diversity Index (Pi)] and, Neutrality tests (Tajima's D and Fu's Fs) were determined using DNA Sequences Polymorphism software (dnaSP) (version 5.1.10). Pairwise differences and population structures of each species were evaluated by analysis of molecular variance (AMOVA) in Arlequin software (version 3.11) (cmpg.unibe.ch/software/arlequin3/) and significance was evaluated based on 1000 permutations. Median joining (MJ) network analysis was conducted based on the number of nucleotide differences and haplotype networks of these three regions, and haplotype networks were constructed using Network software 5.0.0.1 (http://www.fluxus-engineering.com) to determine the interrelationship between haplotypes.

# Results

## Morphological identification

A total of 126 of adult sand flies were collected and were identified as *Phlebotomus argentipes* [15,24,25] and then subjected to PCR amplification followed by DNA sequencing.

## Phylogenetic analysis

A minimum of 10 good quality sequences of mtDNA *cox 1* (~650bp), *cytb* (~550bp) genes and rDNA ITS2 ( ~350bp) region of *P. argentipes* specimens collected from each locality were used in the phylogenetic analysis. Alignment of *cox 1*, *cytb* and ITS2 sequences with those available in public domain in Genbank confirmed their identification as *P. argentipes* (Nucleotide identity: 99.57% for *cox 1* gene, 100.00% for *cytb* gene and 94.15% for ITS2 region) (S1–S3 Tables).

The evolutionary analysis by maximum likelihood method respectively for *cox 1*, *cytb* and ITS2 sequences of *P. argentipes* are shown in Fig 2. Representative haplotypes obtained for sand flies collected from different study sites were included in a single clade. There was no separation of clades based on the study sites suggesting the variations within this species complex and characteristics of individual sand flies are shared among studied populations.

The phylogenetic tree obtained for *cox 1*, *cytb* genes and ITS2 region were separated in to 2 different clades except SLITSI42 sample from Thalawa population (Fig 2A–2C). Each clade of *cox 1* gene (A and B) comprised 21 individuals (29–85% bootstrap value) (Fig 2A) and the clade of *cytb* gene (X and Y) comprised 35 and 7 individuals (Fig 2B) in each clade. The two different clades (P and Q) of ITS2 region respectively comprised 34 and 7 individuals (48–88% bootstrap value) (Fig 2C).

Sand Flies originated from clade A (18 sand flies) of *cox 1* gene clustered with *P. argentipes* previously reported in Sri Lanka and 2 individual sequences from Mirigama and Thalawa from the same clade A grouped with *P. argentipes* found in Kerala, India. Clade B of the *cox 1* gene clustered with *P. argentipes* found in Israel (Fig 2A). Sequences (3 sand flies from Pannala and 1 sand fly from Thalawa) from clade Y of the *cytb* tree grouped with *P. argentipes* sibling sp. A and B previously reported in Sri Lanka and 3 individual sequences from Pannala from the same clade Y grouped with *P. argentipes* found in Madagaskar and nothing from clade X clustered with any publicly available sequence in NCBI (Fig 2B). None of the sequences from clade P and Q of ITS2 region grouped with publicly available sequences in NCBI (Fig 2C).

## DNA sequence polymorphism

Respectively 16, 6 and 3 unique haplotypes were identified with a haplotype diversity (Hd) of 0.878, 0.513 and 0.324 for mtDNA *cox 1* (~650bp), *cytb* (~550bp) genes and rDNA ITS2 ( ~350bp) region respectively. The total number of polymorphic sites were 263, 105 and 83 respectively for *cox 1*, *cytb* genes and ITS2 region. The haplotype diversities (Hd) and nucleotide diversities (Pi) were similarly high for all species in each study site for *cox 1* gene. *P. argentipes* from Mamadala population had the lowest haplotype diversity (0.182) and nucleotide diversity (0.007) with haplotype diversities ranging from 0.182 to 0.733 and nucleotide diversities from 0.007 to 0.276 when compared with data from other populations for *cytb* gene. Similar haplotype diversities values were observed for *cytb* gene and ITS2 region in both Mamadala and Thalawa (Table 1).

According to neutrality test results of *cox 1* gene, both Tajima's D and Fu's Fs values were significant in Mamadala (P<0.02), Thalawa (P<0.05) and Mirigama (P<0.05) populations whereas these values were not significant in Pannala (P>0.1) population (Table 1).

Tajima's D index was positive (+) for all study sites of *cox 1* gene that may indicates a bottleneck situation except in Pannala population. Both Tajima's D and Fu's Fs values in *cytb* gene were not significant (P>0.1) in Mirigama population whereas significant in other populations (P<0.05) where the population may have experienced a bottleneck event in the past (Table 1). When consider neutrality results of ITS2 region, both Tajima's D and Fu's Fs values were significant only in Mamadala population (P<0.02) where the population might have experienced a bottleneck event in the past (Table 1).

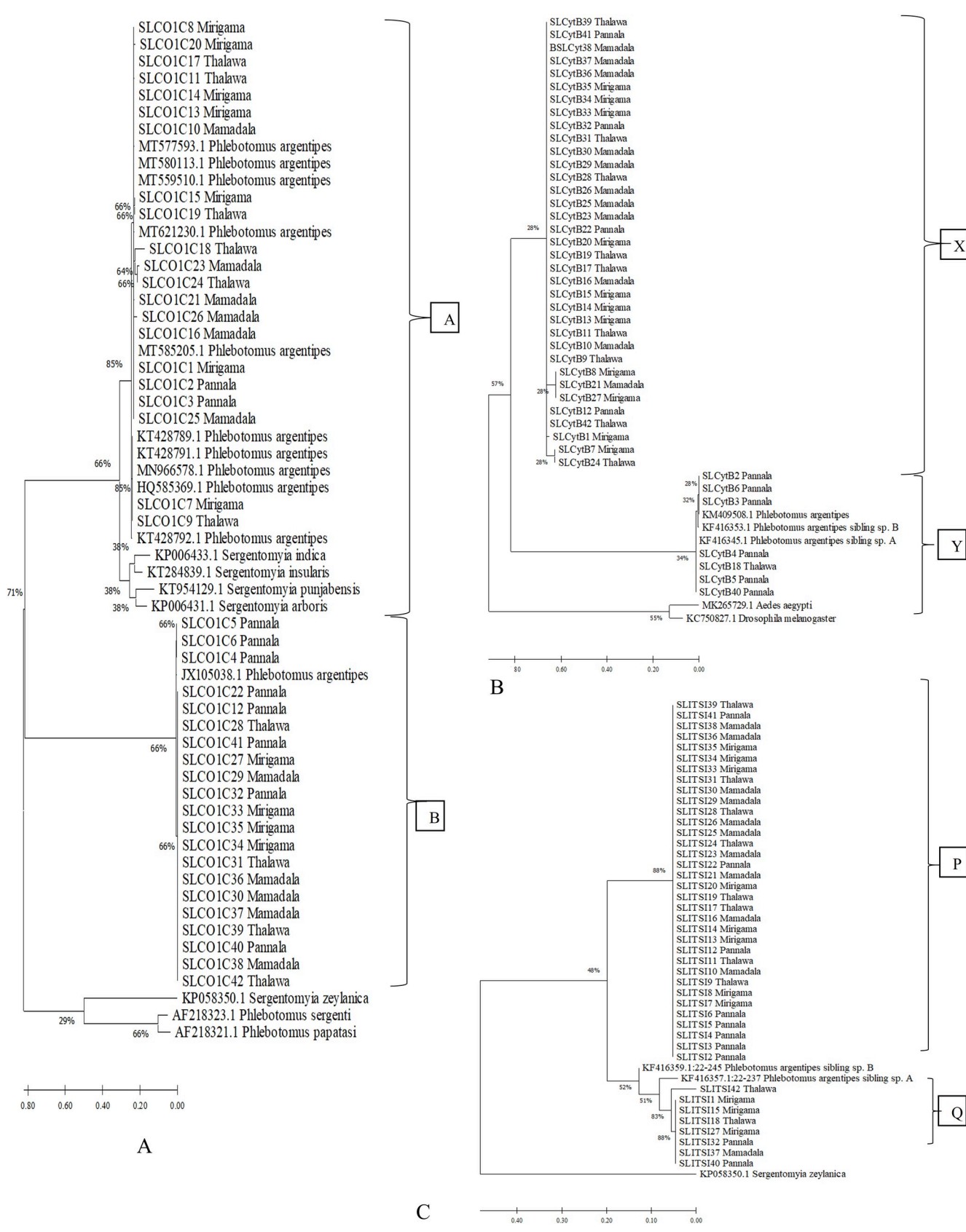

**Fig 2. Evolutionary analysis.** Phylogenetic analysis by Maximum Likelihood method for A)*cox1* B)*cytb* and C)ITS2 region of *P. argentipes* collected from four localities; Mamadala, Mirigama, Pannala, Thalawa.

**Pairwise $F_{ST}$ values and population structure.**  The pairwise $F_{ST}$ values obtained for three different genes of *P. argentipes* in this study are presented in Table 2. The pairwise $F_{ST}$ values of *P. argentipes* between different study sites were not significant (P>0.1) with an absence of genetic differentiation between populations. Interestingly, the pairwise $F_{ST}$ values between Mirigama and Pannala (P = 0.01) and between Mamadala and Pannala (P = 0.00) for *cytb* gene were significant favouring existence of genetic differentiation between populations (S4 Table).

**The effects of evolutionary forces on population.**  $F_{ST}$ (Fixation index) values were 0.0579, 0.4221 and 0.0607 for *cox 1*, *cytb* genes and ITS2 region respectively for *P. argentipes* in this study (Table 3). According to the AMOVA analysis, variations of population genetic structure was not observed among or between populations for *cox 1* gene and ITS2 region whereas significant for (P<0.05) *cytb* gene among studied localities (S4 Table).

## Haplotypes interrelationship

The lowest number of haplotypes (n = 3) were shown by ITS2 region when compared to the *cytb* (6 haplotypes) and *cox 1* (8 haplotypes) genes of *P. argentipes*. The most dominant haplotypes were haplotype 5 and 6 for *cox 1* and *cytb* genes respectively (42.86% and 69.05% of the total population in *cox 1* and *cytb* genes respectively), and haplotype 2 (80.95% of the total population) in ITS2 region (Fig 3A–3C). The lowest haplotype sharing was shown by *cox 1* gene where only 36.36% (haplotypes 1, 3, 4 and 5) were observed while 66.67% haplotypes sharing were shown by both *cytb* gene (haplotypes 3, 4, 5 and 6) and ITS2 (haplotypes 1 and 2). Out of shared haplotypes, only haplotype 5 and 6 respectively for *cox 1* gene and *cytb* gene shared among 4 localities while other haplotypes shared among 2 or 3 study localities (Fig 3A(i) and 3B(i)). However, both haplotypes 1 and 2 in ITS2 region shared among 4 localities in this study (Fig 3C(i)).

**Table 1. Genetic diversity indices, neutrality test values for *Cox 1*, *Cytb* and ITS2 regions of *P. argentipes* in each study site (n≤ 10 from each study site).**

| Population | | n | S | H | Hd (±SD) | Pi | DT | DD | DF |
|---|---|---|---|---|---|---|---|---|---|
| Whole population | *Cox 1* | 42 | 263 | 16 | 0.878±0.037 | 0.427 | -3.150 | -0.526 | -0.464 |
| | *Cytb* | 42 | 105 | 6 | 0.513±0.089 | 0.154 | 0.392 | 1.913 | 1.626 |
| | ITS2 | 42 | 83 | 3 | 0.324±0.081 | 0.074 | 0.583 | 0.656 | 0.750 |
| Mamadala | *Cox 1* | 11 | 252 | 7 | 0.818±0.119 | 0.304 | 2.659 | 1.524 | 2.064 |
| | *Cytb* | 11 | 8 | 2 | 0.182±0.144 | 0.007 | -1.934 | -2.320 | -2.511 |
| | ITS2 | 11 | 76 | 2 | 0.182±0.144 | 0.042 | -2.230 | -2.745 | -2.967 |
| Mirigama | *Cox 1* | 11 | 251 | 6 | 0.836±0.089 | 0.290 | 2.262 | 1.609 | 2.020 |
| | *Cytb* | 11 | 15 | 4 | 0.600±0.154 | 0.022 | -1.331 | -0.737 | -1.007 |
| | ITS2 | 11 | 76 | 2 | 0.436±0.133 | 0.101 | 1.326 | 1.651 | 1.780 |
| Pannala | *Cox 1* | 10 | 253 | 3 | 0.689±0.104 | 0.202 | 2.019 | 1.672 | 1.430 |
| | *Cytb* | 10 | 100 | 3 | 0.733±0.076 | 0.276 | 2.430 | 1.656 | 2.091 |
| | ITS2 | 10 | 76 | 2 | 0.356±0.159 | 0.082 | 0.029 | 1.647 | 1.403 |
| Thalawa | *Cox 1* | 10 | 262 | 7 | 0.867±0.107 | 0.304 | 2.013 | 1.241 | 1.622 |
| | *Cytb* | 10 | 103 | 3 | 0.378±0.181 | 0.109 | -2.141 | -2.522 | -2.743 |
| | ITS2 | 10 | 83 | 3 | 0.378±0.181 | 0.087 | 0.510 | 0.636 | 0.395 |

n, number of DNA sequences for each gene and location; S, number of polymorphic sites; H, number of haplotypes; Hd, haplotype diversity; Pi, nucleotide diversity; DT, Tajima's D; DD, Fu & Li's D*; DF, Fu & Li's F*; SD, Standard deviation.

**Table 2. Pairwise $F_{ST}$ values and P values obtained for *Cox 1*, *Cytb* and ITS2 regions of *P. argentipes* collected in 4 different localities.**

| Study site | Mirigama | Mamadala | Thalawa | Pannala |
|---|---|---|---|---|
| *Cox 1* region | | | | |
| Mirigama | 0.00000 | | | |
| Mamadala | 0.08011 | 0.00000 | | |
| Thalawa | 0.09859 | 0.09670 | 0.00000 | |
| Pannala | 0.24943 | 0.13873 | 0.19531 | 0.00000 |
| *Cytb* region | | | | |
| Mirigama | 0.00000 | | | |
| Mamadala | 0.03186 | 0.00000 | | |
| Thalawa | 0.00144 | 0.00711 | 0.00000 | |
| Pannala | 0.53985* | 0.55809* | 0.35835 | 0.00000 |
| ITS2 region | | | | |
| Mirigama | 0.00000 | | | |
| Mamadala | 0.01579 | 0.00000 | | |
| Thalawa | 0.08035 | 0.05162 | 0.00000 | |
| Pannala | 0.08952 | 0.05307 | 0.10333 | 0.00000 |

* Significantly different pairwise $F_{ST}$ values (p<0.02).

With reference to the *cox 1* gene of *P. argentipes*, haplotype 6, 7 and 8 respectively had a relationship with sand flies isolated previously from Sri Lanka, India and Israel (Fig 3A(ii)). Haplotype 1 (sand flies from Pannala) and 2 (sand flies from Pannala and Thalawa) in *Cytb* region of *P. argentipes* were related to sand flies originated from Madagascar and from Sri Lanka respectively (Fig 3B(ii)). None of the sequences of sand flies shared haplotypes with other reference sand fly sequences in NCBI and therefore ITS2 region of sand flies were considered as unique (Fig 3C(ii)).

## Discussion

Leishmaniasis transmission may be curtailed by the introduction of vector control strategies in Sri Lanka. Previous work by authors demonstrated the existence of insecticide tolerance in local sand fly populations and also the presence of a known genetic mutation associated with insecticide resistance, with direct implications on future sand fly control [10]. Thus, the present study was conducted to investigate the genetic diversity and population genetic structure of leishmaniasis vectors, which are applicable for effective planning of future vector control programmes in Sri Lanka.

This is the first attempt, as far as it is known, to study the population genetic structure of *P. argentipes* using multiple markers to understand the possible evolutionary forces that may operate in local sand fly populations.

**Table 3. $F_{ST}$ values obtained for AMOVA analysis of *Cox 1*, *Cytb* and ITS2 regions of *P. argentipes* collected in each study site.**

| Region | $F_{ST}$ values | P value |
|---|---|---|
| *Cox 1* | 0.0579 | 0.1964±0.0121 |
| *Cytb* | 0.4221* | 0.0020±0.0014 |
| ITS2 | 0.0607 | 0.7801±0.0142 |

* Significantly different $F_{ST}$ values (p = 0.02).

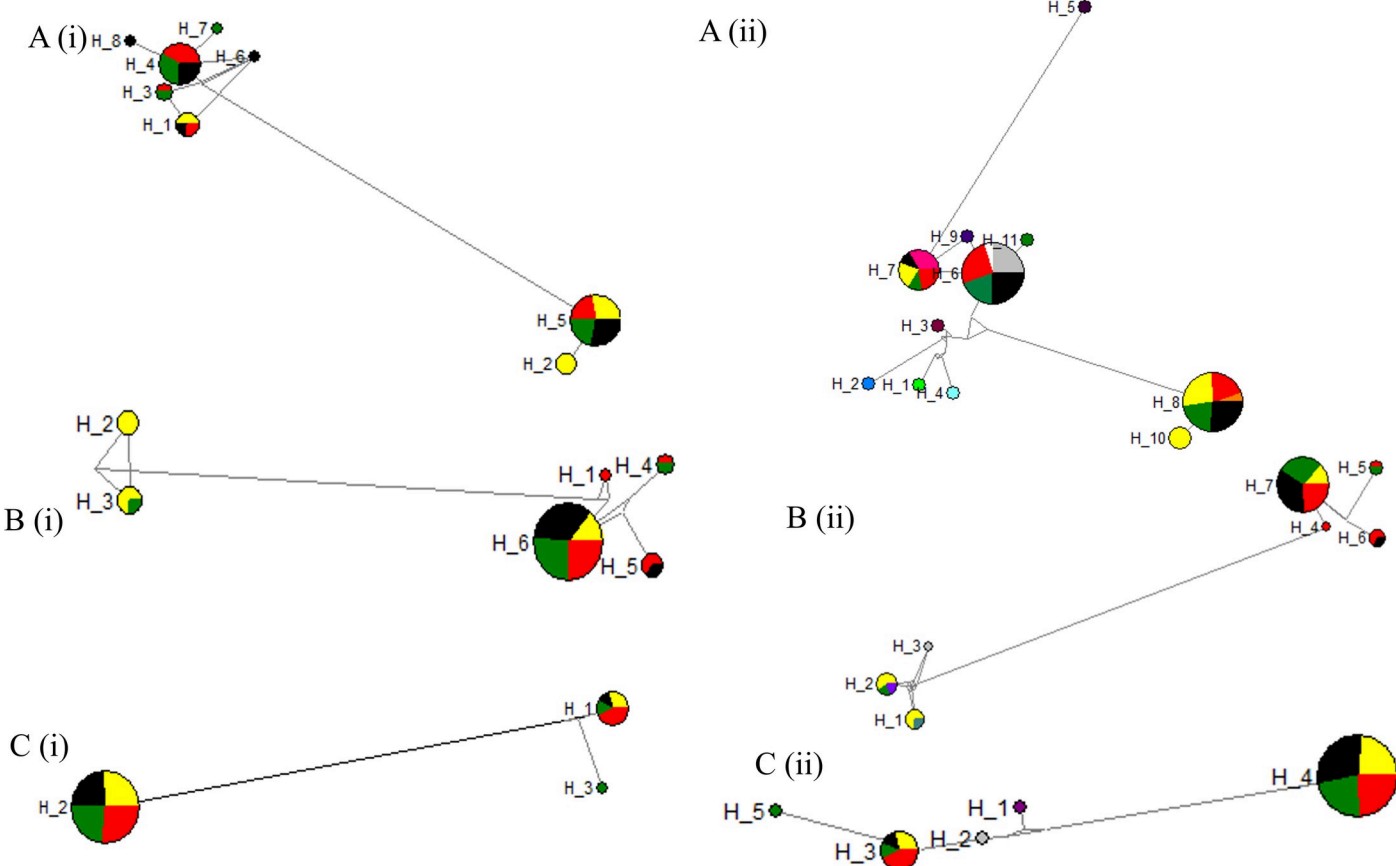

**Fig 3. Haplotypes interrelationship.** Haplotype networks generated using Network 5.0.0.1 for A)*Cox 1* B)*Cytb* C)ITS2 region of *P. argentipes* collected from four geographical locations in Sri Lanka. i) for total population ii) Comparison with NCBI references. Each haplotype is represented by a circle and the size of the circle is proportional to the number of individuals with each haplotype. Geographical localities are colour coded. Mirigama; red, Pannala; yellow, Mamadala; black, Thalawa; green. Reference sequences are indicated in different colours.

The phylogenetic trees generated in the study demonstrated two clades. Interestingly, most of the haplotypes were shared among four locations, which indicate the close genetic relatedness between sand fly populations in the four regions studied.

Phylogenetic tree analysis and network analysis of *cox 1* sequences of this study also indicated that the *P. argentipes* were closely related to specimens of the same species collected in Sri Lanka, India and Israel. However, the *cytb* sequences of three sand flies originated from Pannala were closely related to *P. argentipes* found in Madagaskar, which may need further confirmation with a larger sample. The ITS2 region analysis indicated unique sequences that did not show any resemblance to existing sequences available in the NCBI database.

This is the first attempt to obtained sequences and population genetic analysis for *cox 1*, *cytb* genes and ITS2 region for *P. argentipes* in Sri Lanka.

Tajima's D is the difference between observed average pairwise difference (π) and expected average pairwise difference (θ) [28]. Tajima's D index of *cox 1* gene was positive (+) in all localities i.e. Thalawa, Mirigama and Mamadala, (except in Pannala population) that demonstrates the presence of high variations that exist within study sites with observed average pairwise differences (π) being higher than the expected average pairwise differences (θ). Thus the natural selection may have led to a low variation, which might result in a bottleneck event. This effect was further confirmed using ITS2 region analysis, which resulted in a positive (+) Tajima's D value in Pannala.

Tajima's D of *cytb* gene was minus (-) for Thalawa and Mamadala indicating a low genetic variation within the study sites and therefore, the π is lower than θ. This might indicate a bottleneck event in the past in these sites. Such an effect could be confirmed in Mamadala also based on ITS2 region analysis.

High $F_{ST}$ value of 0.4221 with *cytb* gene analysis in this study suggest genetically distinct populations between the sites (P<0.02). However, that assumption does not hold true when $F_{ST}$ values from *cox 1* gene and ITS2 region were considered (P>0.05). Interestingly, the pairwise $F_{ST}$ values between Mirigama and Pannala and between Mamadala and Pannala for *cytb* gene were different (P<0.02), which may demonstrate genetic differentiation between populations.

The lack of population differentiation and haplotype sharing between the sites that was revealed in the population structure analysis (both pairwise comparisons and $F_{ST}$ values derived through the analysis of molecular variance) using *cox 1* and ITS2 markers may indicate a possible gene flow between regions (though it's not supported by the outcome of *cytb* region analysis in Mamadala, Mirigama and Pannala). Sri Lanka is an island with a relatively small land mass and there are no major geographical barriers between the studied localities. Hence, regardless of the geographic distance the likelihood of a gene flow between the study sites could be considerably high. The presence of a known genetic mutation associated with insecticide resistance in a notable proportion of flies in these study locations was previously reported [10]. A possible gene flow therefore, would pose a risk of dispersal of genes of significance, such as insecticide resistance genes, between populations with the risk of spread of insecticide resistance that will have implications for future vector control. The findings related to gene flow could be further strengthened through future studies involving a larger sample size.

The molecular markers have been extensively used for genotyping and study of evolutionary history of mosquitoes. The analysis using *cytb* gene has demonstrated to be useful to detect the lack of population differentiation and the presence of gene flow in some species of dipterans in South-east Asia [29]. The gene flow between species of insect vectors may lead to alterations in the disease patterns [30]. Low genetic structure variation in some sand fly species as a result of geographical isolation and restricted gene flow may have led to reduced flight abilities and formation of cryptic species that ultimately influence the capacity to transmit parasites such as *Leishmania* in Latin America [31]. Moreover, *cytb* investigations have demonstrated the high genetic differentiation and restricted gene flow among populations of *Lutzomyia cruciata* in Mexico [32]. The low intraspecific genetic divergence between specimens from Mexico and those from north-western Colombia of *Lu. shannoni* population may be interpreted as a northward expansion of the species from South America through Central America via the Isthmus of Panama. Interestingly, a recent population expansion of *Lu. shannoni* in the U.S coincides with the conclusion of the most recent great glaciation period at the end of the Pleistocene Epoch [unpublished work].

Population genetic structure studies of sand fly vectors is important to investigate the presence of possible gene flow among populations. The possible gene flow might act as a carrier and transfer genes from one population to another even in geographically distinct populations (devoid of geographical barriers). If the genes are responsible for insecticide resistance, the gene flow might enhance the number of species having insecticide resistance genes with adverse consequences on vector control programs. Thus it is vital to investigate the population genetic structure of sand flies in Sri Lanka to enable effective sand fly control programs in the future.

## Conclusions

Populations of *P. argentipes* in Sri Lanka seem to be distinct from sand flies that are found elsewhere except for their sequences related to *cox 1* gene (100%) and *cytb* gene (16.7%) that were

found to be shared between study populations. Furthermore, the findings related to the analyses of both *cox 1* and ITS2 indicated a possible gene flow with lack of population differentiation between geographically distant populations of *P. argentipes* in Sri Lanka, perhaps due to the absence of geographic barriers and the continuity of habitats. Therefore, there is a high risk of transferring genes of functional significance, such as those that confer insecticide resistance from one population to another with the risk of eventually spreading across the country. However, confirmation of such an assumption would require further investigations and supportive evidence probably with the use of more neutral genotyping markers such as microsatellites, which have been used in other *Phlebotomus* sp. for similar investigations including for those related to the influence of geographic distance and altitude [33].

The study also confirms the possible use of *cox 1*, *cytb* genes and ITS2 region as genotyping markers for the determination of the population genetic structure of Phlebotominae sand flies in Sri Lanka. Thus, the findings will pave the way for more extensive investigations, probably with the use of more genotyping markers, to aid a vector control strategy that may need to be adopted within a future national leishmaniasis control programme.

## Supporting information

**S1 Table. Nucleotide identities for *cox 1* gene.** Nucleotide identities obtained through NCBI blast results for *cox 1* gene.
(PDF)

**S2 Table. Nucleotide identities for *cytb* gene.** Nucleotide identities obtained through NCBI blast results for *cytb* gene.
(PDF)

**S3 Table. Nucleotide identities for ITS2 region.** Nucleotide identities obtained through NCBI blast results for ITS2 region.
(PDF)

**S4 Table. Analysis of Molecular Variance (AMOVA) of *cox 1*, *cytb* genes and ITS2 region.** Analysis of Molecular Variance (AMOVA) of *cox 1*, *cytb* genes and ITS2 region using Arlequin software (version 3.11) (cmpg.unibe.ch/software/arlequin3/).
(PDF)

## Acknowledgments

We acknowledge Mr. Sunil Shantha and Mr. M. P. Ariyapala for field assistance to collect sand flies, Mr. Wasantha Senadeera for providing a map of study locations, the Head and staff of the Parasitic Diseases Research Unit, Department of Parasitology, Faculty of Medicine, University of Colombo for logistical help.

## Author Contributions

**Conceptualization:** S. H. P. Parakrama Karunaratne, Nadira D. Karunaweera.

**Data curation:** Dulani Ruwanika K. Pathirage, Nadira D. Karunaweera.

**Formal analysis:** Dulani Ruwanika K. Pathirage, Thilini C. Weeraratne, S. H. P. Parakrama Karunaratne.

**Investigation:** Dulani Ruwanika K. Pathirage.

**Methodology:** Dulani Ruwanika K. Pathirage, Sanath C. Senanayake, S. H. P. Parakrama Karunaratne, Nadira D. Karunaweera.

**Project administration:** Nadira D. Karunaweera.

**Resources:** Sanath C. Senanayake, S. H. P. Parakrama Karunaratne, Nadira D. Karunaweera.

**Writing – original draft:** Dulani Ruwanika K. Pathirage.

**Writing – review & editing:** Nadira D. Karunaweera.

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
