## [Decision Letter · Decision Letter 0]

22 Jun 2021

PONE-D-21-16188

Genetic diversity and population structure of Phlebotomus argentipes: vector of leishmaniasis in Sri Lanka.

PLOS ONE

Dear Dr. Karunaweera,

Thank you for submitting your manuscript to PLOS ONE. After careful consideration, we feel that it has merit but does not fully meet PLOS ONE’s publication criteria as it currently stands. Therefore, we invite you to submit a revised version of the manuscript that addresses the points raised during the review process.

We look forward to receiving your revised manuscript.

Kind regards,

Maria Stefania Latrofa

Academic Editor

PLOS ONE

Additional Editor Comments:

Dear Authors, thought the study is of interest for the readers, it is pivotal to modify the ms. according to comments of both reviewers and focusing the population genetic analysis on microsatellite loci; otherwise, you have to limit the study only on the molecular typing of sand fly species examined.

Journal Requirements:

2. We note that Figure 1 in your submission contain map images which may be copyrighted. PLOS content is published under the Creative Commons Attribution License (CC BY 4.0), which means that the manuscript, images, and Supporting Information files will be freely available online, and any third party is permitted to access, download, copy, distribute, and use these materials in any way, even commercially, with proper attribution. For these reasons, we cannot publish previously copyrighted maps or satellite images created using proprietary data, such as Google software (Google Maps, Street View, and Earth). For more information, see our copyright guidelines: http://journals.plos.org/plosone/s/licenses-and-copyright.

2.1.    You may seek permission from the original copyright holder of Figure 1 to publish the content specifically under the CC BY 4.0 license. 

2.2.    If you are unable to obtain permission from the original copyright holder to publish these figures under the CC BY 4.0 license or if the copyright holder’s requirements are incompatible with the CC BY 4.0 license, please either i) remove the figure or ii) supply a replacement figure that complies with the CC BY 4.0 license. Please check copyright information on all replacement figures and update the figure caption with source information. If applicable, please specify in the figure caption text when a figure is similar but not identical to the original image and is therefore for illustrative purposes only.

Reviewers' comments:

Reviewer's Responses to Questions

**Comments to the Author**

1. Is the manuscript technically sound, and do the data support the conclusions?

Reviewer #1: No

Reviewer #2: Partly

2. Has the statistical analysis been performed appropriately and rigorously? 

Reviewer #1: No

Reviewer #2: Yes

3. Have the authors made all data underlying the findings in their manuscript fully available?

Reviewer #1: Yes

Reviewer #2: Yes

4. Is the manuscript presented in an intelligible fashion and written in standard English?

Reviewer #1: Yes

Reviewer #2: No

5. Review Comments to the Author

Reviewer #1: Revision of the paper “Genetic diversity and population structure of Phlebotomus argentipes: vector of leishmaniasis in Sri Lanka” by Pathirage et al.

Leishmaniasis is a significant vector-borne disease. Understanding the degree of gene flow among vector populations is essential for planning efficient control actions and avoiding resistant allele spread. At this aim, studying the genetic structure of populations is an important and well-recognized approach.

In this paper the authors aimed to determine the population genetic structure of sand fly vectors in Sri Lanka. Two mitochondrial genes (Cox 1 and Cytb) and the internal transcribed spacer (ITS2) region from the nuclear ribosomal DNA were used for molecular characterization.

The markers used in this paper are not suitable for assessing population genetic structure and gene flow among populations. At this aim, microsatellite or SNPs markers should be used. Consequently the inferred conclusions are not supported by genetic data.

The genetic analyses carried out in this paper revealed unique sequences of “all genomic regions studied except the cox 1 gene in 21 flies that aligned with those from Kerala, India and cytb gene of 4 flies that aligned with those isolated earlier from Sri Lanka and 3 from Madagascar”.

The authors say that Cox 1 gene and ITS 2 region analyses revealed gene flow between the study sites”. However, the lack of differentiation among populations is due to incomplete lineage sorting and the lack of power of these markers for intra-specific studies.

Reviewer #2: Genetic diversity and population structure of Phlebotomus argentipes: vector of leishmaniasis in Sri Lanka

General comment:

Kalawila et al. present a very interesting article about the genetic variability of the species Phlebotomus argentipes collected in four areas of Sri Lanka. This species is of great relevance for its role in the transmission of visceral leishmaniasis. They analyzed three genes, two mitochondrial and one ribosomal, with which they observed the presence of two well-differentiated communities in two different clades. Additionally, they analyzed the population structure, finding the gene flow between the analyzed populations. They increase the number of genetic sequences for three genes, but mainly provide the first ITS2 gene sequences for the region and the species.

In general, I consider that it is an article that contains interesting findings, but I suggest some changes that I consider necessary to favor the content of the article. I also suggest that the article be reviewed by an English language consultant.

I wish you much success and I hope that my suggestions will be useful to you. Congratulations for your article.

Particular comments

Abstract

Line 30: Two mitochondrial genes viz. Please correct this sentence, I consider the word “viz.” it is misused.

Line 31: please delete oxidase, just leave Cytochrome b (CytB)

Line 30-31: parentheses are not italicized

Line 33: Change Network to network

Line 35: change flies to sand flies

Line 38-42: Please restructure the sentences, because it is a bit confusing.

Introduction

Line 51-54: I suggest change the paragraph this way:

Phlebotomus argentipes (Diptera: Psycodidae) is the known vector of Leishmania donovani, the causal agent of the clinical form VL which is considered the second parasitic disease cause of a high number of deaths, in India, Nepal and Bangladesh [4].

Line 70: change weren´t to were not

Line 78, 83, 103: delete this space

Line 86: Colombia is not an Old world country, please make the correction. Additionally there are more barcode studies for species from the new world, such as USA, Mexico, Brazil, please keep it in mind

Line 99: change phylogeograpy to phylogeography

Line 61-77: Although the information is relevant is very long, please be more specific.

Line 93-102: I suggest that only provide the information regarding the relevance of the use of the cox1, cytb and ITS2 genes for the genetic characterization of Phlebotomus argentipes. And do not include information that is not relevant in your research. For example: Line 94-96:In South America the phylogenetic analysis of cox 1 sequences of Verrucarum species of New World sand flies of the genus Lutzomyia was investigated [20].Line:97-98: Mitochondrial introgression in the Lutzomyia townsendi in Colombia was studied using cytb gene [12].

Line 48-107: In general, the introduction is very long, and contains information that is not very relevance. I kindly suggest that it be summarized a bit to make it more clear and precise.

Line106-107: “to study the effect of evolutionary forces that may spread resistance genes from one population to another”.

I have a question. The genes to be amplified are not necessarily genes that are related to insecticide resistance. How do you plan to evaluate and/or justify that the genetic variability that you may find are related to resistance to insecticides?

It is not very clear to me. Please in the part of your introduction, where you talk about insecticides could explain what genes are useful for do this objective.

Material and Method

Line 111-116: A total of 126 of adult sand flies were collected, this sentences is a result not is part of the method. Please move this information to the next section. Also specify how many days you collected in each of those locations.

Results

Line:189-190: A total of 126 of adult sand flies were collected and were identified as Phlebotomus argentipes [15, 25, 26].

Please, also specify how many specimens you collected in each one of the four locations that you studied. Why you did not collect other sand fly species and only obtain specimens of Phlebotomus argentipes, this is very common in other studies? There are not other sand fly species in your country? How many males and females did you collect? Did you analysis 126 specimens by PCR or how many?

Line193: how did you select those 10 specimens of the best quality?

Line 214, 217,305,306,308: Change flies or fly to sand flies or sand fly according to the case in the entire document.

Line 217: correct P. argentipes

Line 237: In the table, please include the number of specimens that you analyzed, for each gene and locations.

Discussion

Line 322: two clades

Line 325: network analysis

Line 326: P. argentipes were closely related to specimens of the same species collected in Sri Lanka, India and Israel.

Line 327: of three sand flies

Line 327-328: cytb is a marker that shows variability at the intraespecific level, different from cox1 that shows differences at the interespecific level. So, why is relevant that some specimens are similar to specimens from Madagascar? Explain more about it, although you have a small sample, it is interesting that you have haplotypes similar to a locality that is considerably distant from your collection point.

Line 329: did not

Line 330: Add a sentence highlighting that your work is one of the first sequences for this gene and for this species. I think it is very important that you highlight this contribution of your work.

Line 345: does not.

Line 352-353: put the name of the area of study not numbers, because you did not listed before.

Line 354-360: Although it is an interesting hypothesis, I think it is very risky to assert something of this style, given that the number of specimens that you analyzed is very small. I do not consider that ten specimens are enough to think that there is gene flow in all communities of sand flies and those possible resistance genes are being inherited. Additionally, you did not analyze genetic markers for resistance.

Line 363-373: This information is not informative for this article, I suggest that you focus on the group of sand flies and compare your results with those of other researchers. I suggest you read the article by:

1. Pech-May et al. 2016. Genetic structure and divergence in populations of Lutzomyia cruciata, a phlebotomine sand fly (Diptera: Psychodidae) vector of Leishmania mexicana in southeastern Mexico https://doi.org/10.1016/j.meegid.2013.02.004

2. Wolkoff, 2018. Population structure, demographic history, and environmental niche of the sand fly disease vector Lutzomyia shannoni (Dyar) (Diptera: Psychodidae) in the U.S., Mexico, and Colombia https://scholarworks.uttyler.edu/biology_grad/53/

Line 362-382: I suggest that you change these paragraphs as follows and fill them in with additional information, so that you can keep comparing your results.

The molecular markers have been extensively used for genotyping and study of evolutionary history of mosquitoes [30, 31]. The analysis using cytb gene has demonstrated been useful to detect the lack of population differentiation and the presence of gene flow in some species of dipterans in South-east Asia [32]. The gene flow between species of insect vectors may lead to alterations in the disease patterns [35]. Low genetic structure variation in some sand fly species as a result of geographical isolation and restricted gene flow may have led to reduced flight abilities and formation of cryptic species that ultimately influence the capacity to transmit parasites such as Leishmania in Latin America [36].

Line 384-391: again, I do not agree with these statements, I suggest that you approach it in another way for greater support, or that you only suggest it as a possible hypothesis in a more subtle way.

References

Please check the structure of the citations again, to make it homogeneous. Scientific and gene names are also missing in italics, and there are spelling mistakes (e.g. Line 4,5,12, 16, 17, 26, 27, 38, 39)

6. PLOS authors have the option to publish the peer review history of their article (what does this mean?). If published, this will include your full peer review and any attached files.

Reviewer #1: No

Reviewer #2: No

---

## [Author Response · Author response to Decision Letter 0]

3 Jul 2021

Author responses to editor’s and reviewers comments:

Additional Editor’s Comments:

Q: it is pivotal to modify the ms. according to comments of both reviewers and focusing the population genetic analysis on microsatellite loci; otherwise, you have to limit the study only on the molecular typing of sand fly species examined.

R: The current study was designed to determine the population genetic structure of sand flies using two mitochondrial markers cox 1, cytb and a ribosomal region ITS2. However, no microsatellite loci were used for genotyping due their unavailability of such markers in P. argentipes. While we as a group, continue our efforts in developing such robust markers for P. argentipes, the authors believe the selected antigenic loci of P. argentipes are good enough for the said purpose due to the following reasons: 

Previous studies have successfully used cox 1and cyb as the genetic marker in analysing the population genetic structure of insects including sand flies. 

Ebrahimi, et al. Genetic dynamics in the sand fly (Diptera: Psychodidae) nuclear and mitochondrial genotypes: evidence for vector adaptation at the border of Iran with Iraq. Parasites Vectors 9, 319 (2016). https://doi.org/10.1186/s13071-016-1603-5.

Flanley, et al. Population genetics analysis of Phlebotomus papatasi sand flies from Egypt and Jordan based on mitochondrial cytochrome b haplotypes. Parasites Vectors. 2018; 11, 214, doi:10.1186/s13071-018-2785-9. 

Guernaoui, et al. Population Genetics of Phlebotomus papatasi from Endemic and Nonendemic Areas for Zoonotic Cutaneous Leishmaniasis in Morocco, as Revealed by Cytochrome Oxidase Gene Subunit I Sequencing. Microorganisms. 2020; 8(7): 1010.

Arrivillaga et al. Phylogeography of the neotropical sand fly Lutzomyia longipalpis inferred from mitochondrial DNA sequences. Infect Genet Evol. 2002; 2: 83-95.

Latrofaa et al. Multilocus molecular and phylogenetic analysis of phlebotomine sand flies ( Diptera : Psychodidae ) from southern Italy. Acta trop. 2011;119:91–8. doi:10.1016/j.actatropica.2011.04.013.

Cohnstaedt et al. Phylogenetics of the phlebotomine sand fly group Verrucarum (Diptera: Psychodidae: Lutzomyia). Am J Trop Med Hyg. 2011;84(6):913–22. 

Weeraratne et al. Genetic diversity and population structure of malaria vector mosquitoes Anopheles subpictus, Anopheles peditaeniatus, and Anopheles vagus in five districts of Sri Lanka. Malar. J. 2018; 17: 271. https://doi.org/10.1186/s12936-018-2419-x.

Gene flow also successfully determined using these markers in following studies.

Marcondes et al. Introgression between Lutzomyia intermedia and both Lu. neivai and Lu. whitmani, and their roles as vectors of Leishmania braziliensis. Trans R Soc Trop Med Hyg. 1997; 91(6): 725-6. 

Belen et al. Genetic structures of sand fly (Diptera: Psychodidae) populations in a leishmaniasis endemic region of Turkey. Journal of vector biology. 2011; 36 (Supplement 1): 32-48. https://doi.org/10.1111/j.1948-7134.2011.00110.x.

Studies appear below have proven that ITS regions could successfully determine the population genetic structure of many organisms including parasites and mosquitoes. Further, Manni et al (2015) have used microsatellites and ITS2 as the genetic markers and have proven that both markers are equally successful in determining the population genetic structure.

Manni, M., Gomulski, L.M., Aketarawong, N. et al. Molecular markers for analyses of intraspecific genetic diversity in the Asian Tiger mosquito, Aedes albopictus . Parasites Vectors 8, 188 (2015). https://doi.org/10.1186/s13071-015-0794-5

Villalobos, G., Orozco-Mosqueda, G.E., Lopez-Perez, M. et al. Suitability of internal transcribed spacers (ITS) as markers for the population genetic structure of Blastocystis spp. Parasites Vectors 7, 461 (2014). https://doi.org/10.1186/s13071-014-0461-2. 

Latrofaa MS, Dantas-Torresa F, Weigla S, Taralloa VD, Parisib A, Traversac D, et al. Multilocus molecular and phylogenetic analysis of phlebotomine sand flies ( Diptera : Psychodidae ) from southern Italy. Acta trop. 2011;119:91–8. doi:10.1016/j.actatropica.2011.04.013.

Journal Requirements:

The manuscript was checked and changed according to the guidelines provided in the PLOS ONE journal and further both above links also were used.

2. We note that Figure 1 in your submission contain map images which may be copyrighted. PLOS content is published under the Creative Commons Attribution License (CC BY 4.0), which means that the manuscript, images, and Supporting Information files will be freely available online, and any third party is permitted to access, download, copy, distribute, and use these materials in any way, even commercially, with proper attribution. For these reasons, we cannot publish previously copyrighted maps or satellite images created using proprietary data, such as Google software (Google Maps, Street View, and Earth). For more information, see our copyright guidelines: http://journals.plos.org/plosone/s/licenses-and-copyright.

2.1. You may seek permission from the original copyright holder of Figure 1 to publish the content specifically under the CC BY 4.0 license. 

In the figure caption of the copyrighted figure, please include the following text: “Reprinted from [ref] under a CC BY license, with permission from [name of publisher], original copyright [original copyright year].

Agreed your kind request and done it. The figure 1 was reprinted from authors’ previous work [Pathirage et al. 2020]. Thus, the written permission from the copyright holder to publish these figures specifically under the CC BY 4.0 license was obtained and upload the completed Content Permission Form and the original figure in the previous work as an “Other” file when submitting the manuscript. Further, in the figure caption “Reprinted from authors’ previous work [Pathirage et al. 2020]” was stated in line 105-106 (page number 5) under Materials and Method section in the manuscript.

 Reviewer Comments:

Reviewer 1:

Q1: Is the manuscript technically sound, and do the data support the conclusions?

Reviewer #1: No

Reviewer #2: Partly

R1: A minimum of 10 sequences of each marker from each study site was used in the analysis. Previous published work on population genetic structure analysis have used sample numbers between 6-10 from a single study location and have obtained successful results. I suggest adding one statement to the discussion in line 342-343 and changing the following sentence in the conclusion in line 370-373, 375-377– 

‘Gene flow was evident with lack of population differentiation even between geographically distant populations of P. argentipes perhaps due to absence of geographic barriers and the continuity of habitats.’ 

‘Results also validated the use of cox 1, cytb genes and ITS2 region as a tool in understanding the population genetic structure of Phlebotominae sand flies in Sri Lanka.’

As, 

Both cox 1 and ITS2 indicated a gene flow with lack of population differentiation even between geographically distant populations of P. argentipes perhaps due to absence of geographic barriers and the continuity of habitats.

Results showed the possible use of cox 1, cytb genes and ITS2 region as markers in determining the population genetic structure of Phlebotominae sand flies in Sri Lanka.

Q2. Has the statistical analysis been performed appropriately and rigorously?

Reviewer #1: No

Reviewer #2: Yes

 R2: Standard statistical software and analysis were used in the study by referring to previous published work. 

Q3: The markers used in this paper are not suitable for assessing population genetic structure and gene flow among populations. At this aim, microsatellite or SNPs markers should be used. Consequently the inferred conclusions are not supported by genetic data.

The genetic analyses carried out in this paper revealed unique sequences of “all genomic regions studied except the cox 1 gene in 21 flies that aligned with those from Kerala, India and cytb gene of 4 flies that aligned with those isolated earlier from Sri Lanka and 3 from Madagascar”.

The authors say that Cox 1 gene and ITS 2 region analyses revealed gene flow between the study sites”. However, the lack of differentiation among populations is due to incomplete lineage sorting and the lack of power of these markers for intra-specific studies.

R3: The current study was designed to determine the population genetic structure of sand flies using two mitochondrial markers cox 1, cytb and a ribosomal region ITS2. However, no microsatellite loci were used for genotyping due their unavailability of such markers in P. argentipes. While we as a group, continue our efforts in developing such robust markers for P. argentipes, the authors believe the selected antigenic loci of P. argentipes are good enough for the said purpose due to the following reasons: 

Most of the studies have been conducted to investigate population genetic analysis using these markers (Ebrahimi et al. 2016, Flanley et al, 2018, Guernaoui et al. 2020, Arrivillaga et al. 2002, Cohnstaedt et al. 2011, Testa et al. 2002, Latrofaa et al. 2011, Weeraratne et al. 2018). 

Gene flow also successfully determined using these markers in following studies.

(Marcondes et al. 1997, Belen et al. 2011). 

Further, several studies have conducted using one or two of the markers used in the present study and microsattelites and have proved that both could be used sucessfully in population genetic structure analysis (Manni et al. 2015, Villalobos et al. 2014). 

we suggest changing the following sentence in the conclusion in line 370-373, 375-377– 

‘Gene flow was evident with lack of population differentiation even between geographically distant populations of P. argentipes perhaps due to absence of geographic barriers and the continuity of habitats.’

‘Results also validated the use of cox 1, cytb genes and ITS2 region as a tool in understanding the population genetic structure of Phlebotominae sand flies in Sri Lanka.’

As, 

Both cox 1 and ITS2 indicated a gene flow with lack of population differentiation even between geographically distant populations of P. argentipes perhaps due to absence of geographic barriers and the continuity of habitats.

Results showed the possible use of cox 1, cytb genes and ITS2 region as markers in determining the population genetic structure of Phlebotominae sand flies in Sri Lanka.

Reviewer 2: 

Q1: Is the manuscript technically sound, and do the data support the conclusions?

Reviewer #1: No

Reviewer #2: Partly

R1: A minimum of 10 sequences of each marker from each study site was used in the analysis. Previous published work on population genetic structure analysis have used sample numbers between 6-10 from a single study location and have obtained successful results. I suggest changing the following sentence in the discussion and conclusion in line 342-343, 370-373, 375-377– 

‘Gene flow was evident with lack of population differentiation even between geographically distant populations of P. argentipes perhaps due to absence of geographic barriers and the continuity of habitats.’ 

‘Results also validated the use of cox 1, cytb genes and ITS2 region as a tool in understanding the population genetic structure of Phlebotominae sand flies in Sri Lanka.’

As, 

Both cox 1 and ITS2 indicated a gene flow with lack of population differentiation even between geographically distant populations of P. argentipes perhaps due to absence of geographic barriers and the continuity of habitats.

Results showed the possible use of cox 1, cytb genes and ITS2 region as markers in determining the population genetic structure of Phlebotominae sand flies in Sri Lanka.

Q2: Is the manuscript presented in an intelligible fashion and written in standard English?

Reviewer #1: Yes

Reviewer #2: No

R2: The typographical and grammatical errors suggested by the reviewers were corrected and changes are highlighted in the manuscript. 

Abstract

Q3: Line 30: Two mitochondrial genes viz. Please correct this sentence, I consider the word “viz.” it is misused.

R3: Agreed and changed the sentence in line 30.

Q4: Line 31: please delete oxidase, just leave Cytochrome b (CytB)

R4: Agreed and deleted it in line 31.

Q5: Line 30-31: parentheses are not italicized

R5: Agreed and done.

Q6: Line 33: Change Network to network

R6: Done it in line 33. 

Q7: Line 35: change flies to sand flies

R7: Done it in line 35.

Q8: Line 38-42: Please restructure the sentences, because it is a bit confusing.

R8: Changed the sentence in line 38-42.

Introduction

Q9: Line 51-54: I suggest change the paragraph this way:

Phlebotomus argentipes (Diptera: Psycodidae) is the known vector of Leishmania donovani, the causal agent of the clinical form VL which is considered the second parasitic disease cause of a high number of deaths, in India, Nepal and Bangladesh [4].

R9: Agreed and done the changes to line 50-52.

Q10: Line 70: change weren´t to were not

R10: Agreed and done in line 65.

Q11: Line 78, 83, 103: delete this space

R11: Deleted spaces.

Q12: Line 86: Colombia is not an Old world country, please make the correction. Additionally there are more barcode studies for species from the new world, such as USA, Mexico, Brazil, please keep it in mind.

Q12: Done the changes to the sentence in line 75-77.

Q13: Line 99: change phylogeograpy to phylogeography

R13: Changed as ‘phylogeography’ in line 85.

Q14: Line 61-77: Although the information is relevant is very long, please be more specific.

R14: Agreed and done in line 59-70.

Q15: Line 93-102: I suggest that only provide the information regarding the relevance of the use of the cox1, cytb and ITS2 genes for the genetic characterization of Phlebotomus argentipes. And do not include information that is not relevant in your research. For example: Line 94-96:In South America the phylogenetic analysis of cox 1 sequences of Verrucarum species of New World sand flies of the genus Lutzomyia was investigated [20].Line:97-98: Mitochondrial introgression in the Lutzomyia townsendi in Colombia was studied using cytb gene [12].

R15: Due to restricted studies in P. argentipes corrected statement in line 83-85.

Q16: Line 48-107: In general, the introduction is very long, and contains information that is not very relevance. I kindly suggest that it be summarized a bit to make it more clear and precise.

R16: The changes were made according to the comments suggested by the reviewers to the introduction to be summarized. 

Q17: Line106-107: “to study the effect of evolutionary forces that may spread resistance genes from one population to another”.

I have a question. The genes to be amplified are not necessarily genes that are related to insecticide resistance. How do you plan to evaluate and/or justify that the genetic variability that you may find are related to resistance to insecticides?

R17: There is no link between the genes used in the study and the insecticide resistance genes. This statement was made as there is a high chance of entering such genes from one populations to another population in the absence of population genetic structure variation. 

Q18: It is not very clear to me. Please in the part of your introduction, where you talk about insecticides could explain what genes are useful for do this objective.

R18: Explained in line 67-69.

Material and Method

Q19: Line 111-116: A total of 126 of adult sand flies were collected, this sentences is a result not is part of the method. Please move this information to the next section. Also specify how many days you collected in each of those locations.

R19: Moved it to the result section and added information to the material and method in line 92-97. 

Results

Q20: Line:189-190: A total of 126 of adult sand flies were collected and were identified as Phlebotomus argentipes [15, 25, 26].

R20: Added it to line 171.

Q21: Please, also specify how many specimens you collected in each one of the four locations that you studied. Why you did not collect other sand fly species and only obtain specimens of Phlebotomus argentipes, this is very common in other studies? There are not other sand fly species in your country? How many males and females did you collect? Did you analysis 126 specimens by PCR or how many?

R21: About 42 specimens were collected in each one of the four locations with a total of 52 females and 74 males. There are Sergentomyia spp. in the country. However, trapping method was basically effective for cattle-baited net trap. Thus, Phlebotomus argetipes were found apart from one or two Sergentomyia spp. specimens collected from light traps in some days, not routinely. Due to absence of sufficient numbers of samples, Sergentomyia spp. was not included in the study. Due to the study were based on the vector species, Phlebotomus argentipes, a total of 126 species were analysed by PCR followed by the analysis and interpretation were based on vector control strategies.

Q22: Line193: how did you select those 10 specimens of the best quality?

R22: The chromatograms of each and every sequence was checked manually and only the good quality sequences were used in the study.

Q23: Line 214, 217,305,306,308: Change flies or fly to sand flies or sand fly according to the : case in the entire document.

R23: Agreed and corrected.

Q24: Line 217: correct P. argentipes

R24: Corrected it line 199.

Q25: Line 237: In the table, please include the number of specimens that you analyzed, for each gene and locations.

R25: Added to the Table 1.

Discussion

Q26: Line 322: two clades

R26: Agreed and done it in line 302.

Q27: Line 325: network analysis

R27: Done it in line 305.

Q28: Line 326: P. argentipes were closely related to specimens of the same species collected in Sri Lanka, India and Israel.

R28: Corrected it in line 306-307.

Q29: Line 327: of three sand flies

R29: Corrected it in line 307.

Q30: Line 327-328: cytb is a marker that shows variability at the intraespecific level, different from cox1 that shows differences at the interespecific level. So, why is relevant that some specimens are similar to specimens from Madagascar? Explain more about it, although you have a small sample, it is interesting that you have haplotypes similar to a locality that is considerably distant from your collection point.

R30: The larger sample will be used for further studies to confirm this specimens exactly from Madagascar or not. If so, there is a genetic exchange may apply in between those specimens.

Q31: Line 329: did not

R31: Corrected it in line 309.

Q32: Line 330: Add a sentence highlighting that your work is one of the first sequences for this gene and for this species. I think it is very important that you highlight this contribution of your work.

R32: Done it in line 311-312.

Q33: Line 345: does not.

R33: Corrected it in line 327.

Q34: Line 352-353: put the name of the area of study not numbers, because you did not listed before.

R34: Agreed it and corrected in line 334-335.

Q35: Line 354-360: Although it is an interesting hypothesis, I think it is very risky to assert something of this style, given that the number of specimens that you analyzed is very small. I do not consider that ten specimens are enough to think that there is gene flow in all communities of sand flies and those possible resistance genes are being inherited. Additionally, you did not analyze genetic markers for resistance.

R35: Previously published work on population genetic structure analysis have used sample numbers between 6-10 from a single study location and have obtained successful results. This study used a minimum of 10 sequences from each marker from each locality. However, as the sample size is not very high it was stated as “possible” gene flow. 

Insecticide resistance genes was taken as an example gene that could be introduced into other populations due to gene flow. 

The sentence, “The possible gene flow revealed in the current study demonstrate the risk of dispersal of such resistance genes between populations with the potential for the spread of insecticide resistance and implications for future vector control” could be changed as “The possible gene flow revealed in the current study demonstrate the risk of dispersal of genes such as insecticide resistance genes between populations with the potential for the spread of insecticide resistance and implications for future vector control” in line 339-342 in the discussion.

Q36: Line 363-373: This information is not informative for this article, I suggest that you focus on the group of sand flies and compare your results with those of other researchers. I suggest you read the article by:

1. Pech-May et al. 2016. Genetic structure and divergence in populations of Lutzomyia cruciata, a phlebotomine sand fly (Diptera: Psychodidae) vector of Leishmania mexicana in southeastern Mexico https://doi.org/10.1016/j.meegid.2013.02.004

2. Wolkoff, 2018. Population structure, demographic history, and environmental niche of the sand fly disease vector Lutzomyia shannoni (Dyar) (Diptera: Psychodidae) in the U.S., Mexico, and Colombia https://scholarworks.uttyler.edu/biology_grad/53/

R36: Added information to line 351-358.

Q37: Line 362-382: I suggest that you change these paragraphs as follows and fill them in with additional information, so that you can keep comparing your results.

The molecular markers have been extensively used for genotyping and study of evolutionary history of mosquitoes [30, 31]. The analysis using cytb gene has demonstrated been useful to detect the lack of population differentiation and the presence of gene flow in some species of dipterans in South-east Asia [32]. The gene flow between species of insect vectors may lead to alterations in the disease patterns [35]. Low genetic structure variation in some sand fly species as a result of geographical isolation and restricted gene flow may have led to reduced flight abilities and formation of cryptic species that ultimately influence the capacity to transmit parasites such as Leishmania in Latin America [36].

R37: Added information to line 344-351.

Q38: Line 384-391: again, I do not agree with these statements, I suggest that you approach it in another way for greater support, or that you only suggest it as a possible hypothesis in a more subtle way.

R38: Corrected it in line 360-366.

References

Q39: Please check the structure of the citations again, to make it homogeneous. Scientific and gene names are also missing in italics, and there are spelling mistakes (e.g. Line 4,5,12, 16, 17, 26, 27, 38, 39)

R39: Agreed and done it.

---

## [Editor Report · Decision Letter 1]

4 Aug 2021

PONE-D-21-16188R1

Genetic diversity and population structure of Phlebotomus argentipes: vector of leishmaniasis in Sri Lanka.

PLOS ONE

Dear Dr. Karunaweera,

Thank you for submitting your manuscript to PLOS ONE. After careful consideration, we feel that it has merit but does not fully meet PLOS ONE’s publication criteria as it currently stands. Therefore, we invite you to submit a revised version of the manuscript that addresses the points raised during the review process.

We look forward to receiving your revised manuscript.

Kind regards,

Maria Stefania Latrofa

Academic Editor

PLOS ONE

Journal Requirements:

Additional Editor Comments (if provided):

I agree with the authors on that the target genes analyzed are good for the specimen’s identification and for the analyses of the genetic divergence within and among phlebotomine sand fly populations, but not on the gene flow. Indeed, Weeraratne and colleagues only hypothesized the presence of the gene flow using the cox1, whilst Manni et al., define that SSR markers are, as expected, more informative than ITS2 in revealing the slight genetic diversity between native and derived populations both in terms of variability and differentiation. The authors have to be more cautious when talk on the gene flow. I suggest, for example, putting your considerations as a hypothesis of the “existence of a potential gene flow” based on the results obtained of the genes analyzed, and that the gene flow have to be further investigated or supported by the analysis of microsatellites.

See also the article Prudhomme et al., “Altitude and hillside orientation shapes the population structure of the Leishmania infantum vector Phlebotomus ariasi.” Scientific Reports, 2020 DOI: 10.1038/s41598-020-71319-w.

Change the title: since sand flies are vector of pathogen and not of diseases

Lines 36-37: I suggest changing “revealed gene flow” into “may indicate a potential gene flow”, due to the motivation above, and throughout the ms.

---

## [Author Response · Author response to Decision Letter 1]

16 Aug 2021

Author responses to journal requirements and additional editor’s comments:

Journal Requirements:

Q: Please review your reference list to ensure that it is complete and correct. If you have cited papers that have been retracted, please include the rationale for doing so in the manuscript text, or remove these references and replace them with relevant current references. Any changes to the reference list should be mentioned in the rebuttal letter that accompanies your revised manuscript. If you need to cite a retracted article, indicate the article’s retracted status in the References list and also include a citation and full reference for the retraction notice.

R: Reviewed the reference list and it was completed and corrected. Following changes were made in the reference list.

• Journal name abbreviation was corrected in line 428, 443.

• Page numbers were added in line 467.

• Journal name and page numbers were added in line 473.

• Journal name, issue, page numbers and DOI number were added in line 475-476.

Following reference was removed from the reference list as it contains unpublished work. Thus, the statement in line 353-358 referred to as “unpublished work” in line 358.

 Wolkoff. Population structure, demographic history, and environmental niche of the sand fly disease vector Lutzomyia shannoni (Dyar) (Diptera: Psychodidae) in the U.S., Mexico, and Colombia. University of Texas at Tyler; 2018.

Additional Editor’s Comments:

Q1: I agree with the authors on that the target genes analyzed are good for the specimen’s identification and for the analyses of the genetic divergence within and among phlebotomine sand fly populations, but not on the gene flow. Indeed, Weeraratne and colleagues only hypothesized the presence of the gene flow using the cox1, whilst Manni et al., define that SSR markers are, as expected, more informative than ITS2 in revealing the slight genetic diversity between native and derived populations both in terms of variability and differentiation. The authors have to be more cautious when talk on the gene flow. I suggest, for example, putting your considerations as a hypothesis of the “existence of a potential gene flow” based on the results obtained of the genes analyzed, and that the gene flow have to be further investigated or supported by the analysis of microsatellites.

See also the article Prudhomme et al., “Altitude and hillside orientation shapes the population structure of the Leishmania infantum vector Phlebotomus ariasi.” Scientific Reports, 2020 DOI: 10.1038/s41598-020-71319-w.

R1: Authors agreed and suggested changes have been added in the abstract section in line 32-42 and in the conclusion section in line 369-373 and 374-385.

“Therefore, there is a high risk of transferring genes of functional significance, such as those that confer insecticide resistance from one population to another with the risk of eventually spreading across the country. However, confirmation of such an assumption would require further investigations and supportive evidence probably with the use of more neutral genotyping markers such as microsatellites, which have been used in other Phlebotomus sp. for similar investigations including for those related to the influence of geographic distance and altitude (Prudhomme et al., 2020). The study also confirms the possible use of cox 1, cytb genes and ITS2 region as genotyping markers for the determination of the population genetic structure of Phlebotominae sand flies in Sri Lanka. Thus, the findings will pave the way for more extensive investigations, probably with the use of more genotyping markers, to aid a vector control strategy that may need to be adopted within a future national leishmaniasis control programme.”

Q2: Change the title: since sand flies are vector of pathogen and not of diseases

R2: The title was changed into ‘Genetic diversity and population structure of Phlebotomus argentipes: vector of Leishmania donovani in Sri Lanka.’

Q3. Lines 36-37: I suggest changing “revealed gene flow” into “may indicate a potential gene flow”, due to the motivation above, and throughout the ms.

R3: Changed as ‘may indicate a possible gene flow’ in line 37, 39-40, 333-334, 337, 339-343, 363-364, 372.

Authors suggested to acknowledge funding source in the Acknowledgements section in line 391-394.

---

## [Editor Report · Decision Letter 2]

17 Aug 2021

Genetic diversity and population structure of Phlebotomus argentipes: vector of Leishmania donovani in Sri Lanka.

PONE-D-21-16188R2

Dear Dr. Karunaweera,

We’re pleased to inform you that your manuscript has been judged scientifically suitable for publication and will be formally accepted for publication once it meets all outstanding technical requirements.

Kind regards,

Maria Stefania Latrofa

Academic Editor

PLOS ONE

---

## [Editor Report · Acceptance letter]

3 Sep 2021

PONE-D-21-16188R2 

Genetic diversity and population structure of *Phlebotomus argentipes*: vector of *Leishmania donovani* in Sri Lanka. 

Dear Dr. Karunaweera:

I'm pleased to inform you that your manuscript has been deemed suitable for publication in PLOS ONE. Congratulations! Your manuscript is now with our production department. 

Kind regards, 

on behalf of

Dr. Maria Stefania Latrofa 

Academic Editor

PLOS ONE